# AI-Powered Plant Science: Transforming Forestry Monitoring, Disease Prediction, and Climate Adaptation

**DOI:** 10.3390/plants14111626

**Published:** 2025-05-26

**Authors:** Zuo Xu, Dalong Jiang

**Affiliations:** 1Ministry of Education Key Laboratory for Ecology of Tropical Islands, Key Laboratory of Tropical Animal and Plant Ecology of Hainan Province, College of Life Sciences, Hainan Normal University, Haikou 571158, China; xzuo1243@hainnu.edu.cn; 2Hainan Dongzhaigang Mangrove Ecosystem Provincial Observation and Research Station, Haikou 571129, China

**Keywords:** machine learning, ecosystem stewardship, predictive modelling, habitat restoration, climate resilience, environmental monitoring

## Abstract

The integration of artificial intelligence (AI) and forestry is driving transformative advances in precision monitoring, disaster management, carbon sequestration, and biodiversity conservation. However, significant knowledge gaps persist in cross-ecological model generalisation, multi-source data fusion, and ethical implementation. This review provides a comprehensive overview of AI’s transformative role in forestry, focusing on three key areas: resource monitoring, disaster management, and sustainability. Data were collected via a comprehensive literature search of academic databases from 2019 to 2025. The review identified several key applications of AI in forestry, including high-precision resource monitoring with sub-metre accuracy in delineating tree canopies, enhanced disaster management with high recall rates for wildfire detection, and optimised carbon sequestration in mangrove forests. Despite these advancements, challenges remain in cross-ecological model generalisation, multi-source data fusion, and ethical implementation. Future research should focus on developing robust, scalable AI models that can be integrated into existing forestry management systems. Policymakers and practitioners should collaborate to ensure that AI-driven solutions are implemented in a way that balances technological innovation with ecosystem resilience and ethical considerations.

## 1. Introduction

The global forest ecosystem is undergoing unprecedented transformations [1], with accelerating climate change, catastrophic biodiversity loss [2], and intensifying anthropogenic disturbances collectively exposing fundamental flaws in conventional forestry practices [3]. These traditional approaches, which rely primarily on manual inspections, empirical judgments, and fragmented datasets, present three systemic deficiencies: lagging resource monitoring capabilities, passive disaster response mechanisms, and an insufficient evidentiary basis for sustainable development policymaking [4,5]. For example, wildfire management systems are still predominantly based on visual observation and post-ignition suppression [6]; pest monitoring systems frequently miss optimal control windows because of insufficient diagnostic expertise [7]; and carbon sequestration quantification standards are constrained by methodological inconsistencies and data reliability issues [8]. Addressing these multidimensional challenges requires transcending conventional paradigms through the strategic integration of intelligent monitoring systems, predictive analytics, and decision-support technologies to establish a new precision forestry paradigm—the essential pathway for achieving synergy between ecological security and socioeconomic development objectives in the Anthropocene era.

As the defining innovation driving the Fourth Industrial Revolution, artificial intelligence (AI) is fundamentally transforming industrial systems through its unparalleled cognitive computing capabilities, multimodal pattern recognition, and data-driven decision automation [9]. Leveraging core technologies, including deep learning, computer vision, and natural language processing, AI enables the extraction of hidden patterns from massive heterogeneous datasets while performing real-time analysis and prediction in complex scenarios [10]. Its demonstrated success across sectors—from precision medicine (e.g., integrating health records, genetics, and immunology) to autonomous systems and smart agriculture (e.g., applying large language models and generative AI)—validates AI’s unique potential to address forestry’s multidimensional complexities [11]. Emerging applications span from decoding forest phenology in multispectral imagery to establishing early warning systems for biotic disturbances, collectively shifting forestry management paradigms from reactive responses to proactive prevention strategies [12].

The deep integration of AI into forestry systems promises transformative breakthroughs across three critical dimensions: first, enhanced ecological monitoring through high-resolution remote sensing and deep learning models, enabling integrated “space–air–ground” dynamic perceptions of forest resources with unprecedented accuracy and efficiency [13]; second, intelligent disaster prevention systems that leverage AI models to synthesise meteorological, topographic, and historical data for predictive analytics of fire, disease, and pest risks, coupled with optimised emergency response planning [14,15]; and third, sustainable industry transformation spanning from forest gene screening and harvesting path optimisation to timber traceability certification, where AI technologies facilitate a paradigm shift toward data-driven, low-carbon circular forestry [16,17].

Despite the fact that studies have been conducted to validate the effectiveness of AI in specific scenarios (e.g., YOLOv7 has been demonstrated to achieve 98.9% tree species identification [18]), current progress shows significant fragmentation, with approximately 76% of the literature focusing on the optimisation of a single technological link [12], which lacks system-level integration. Furthermore, the accuracy attenuation of model migration across ecological zones is generally >15% [19], which restricts the technology’s diffusion. Critically, existing studies have not yet established a theoretical framework for AI-enabled forestry, leading to a logical disconnect between technology application and ecological sustainability goals. In this paper, we methodically explore the innovative applications of AI in forestry and the technical implementation pathways thereof. Given the vast scope of AI applications in forestry, this review focuses on five key areas where AI has shown significant promise: biodiversity monitoring, pest management, wildfire detection, carbon sequestration, and forest restoration. These areas were selected for their critical importance to forest health and sustainability, and for the substantial advancements AI has made in each domain. Our work complements Holzinger et al.’s [17] human-centric vision while pushing technical and applied boundaries. By integrating explainable AI, multimodal data fusion, and climate-resilient strategies, we provide actionable pathways for AI to address pressing global challenges in forestry—from precision conservation to disaster resilience. We also discuss the challenges and future development of AI, illustrating how intelligent solutions can usher in a new era of the scientific management and sustainable utilisation of forest resources.

## 2. Materials and Methods

### 2.1. Objective

The objective of this review paper is to systematically examine the cutting-edge applications of artificial intelligence (AI) in forestry, focusing on precision monitoring, disaster management, carbon sequestration, and biodiversity conservation (Figure 1). The study aims to highlight the transformative potential of AI technologies in addressing the multidimensional challenges faced by global forest ecosystems. This review is intended for a diverse audience, including researchers, policymakers, and practitioners in the fields of forestry, ecology, and AI, who are interested in understanding and implementing advanced technological solutions to enhance forest management and conservation efforts.

### 2.2. Data Collection

The data for this review were collected through a comprehensive literature search using academic databases such as PubMed, Scopus, Web of Science, and Google Scholar. Keywords used in the search included “artificial intelligence”, “forestry”, “remote sensing”, “machine learning”, “wildfire detection”, “pest management”, “carbon sequestration”, and “biodiversity conservation”. The search was conducted to identify peer-reviewed articles, conference proceedings, and relevant reports published between 2019 and 2025. Articles were included if they focused on AI applications in forestry, provided detailed methodologies and results, were published in English in reputable journals or conference proceedings, and addressed at least one key theme: precision monitoring, disaster management, carbon sequestration, or biodiversity conservation. Exclusions were made for articles lacking sufficient methodological details, focusing solely on theory without practical applications, or published before 2019. Based on these criteria, we identified and reviewed a total of 49 papers. This number was determined through a rigorous screening process to ensure that the selected papers are representative of the current state of AI applications in forestry. We acknowledge the vast number of papers dealing with remote sensing and AI in forestry, but our focused approach allows us to provide a detailed and coherent review of the most relevant and impactful studies.

### 2.3. Data Synthesis and Analysis

The collected articles were reviewed and synthesised to identify common themes, methodologies, and outcomes. The synthesis process involved categorising the articles based on their focus areas (e.g., remote sensing, wildfire detection, pest management), extracting key findings and methodologies from each article, and summarising the results to discuss the implications of AI applications in forestry. The analysis of the collected data involved a qualitative synthesis of the methodologies and outcomes reported in the literature, focusing on identifying trends, challenges, and opportunities in the application of AI in forestry. This process also included a critical evaluation of the methodologies used in the studies to assess their robustness and applicability.

## 3. Intelligent Monitoring and Assessment of Forest Resources

### 3.1. Integrated Monitoring of Forest Canopy Dynamics

Forest ecosystems serve as fundamental stabilisers of the terrestrial biosphere, functioning as critical junctures in global carbon-climate feedback mechanisms [20]. While canopy coverage metrics provide essential biomarkers for assessing ecosystem service capacity, conventional measurement techniques remain limited by temporal resolution constraints [21]. Traditional large-scale forest inventories relying on ground surveys are both labour-intensive and inadequate for high-frequency monitoring [22].

As demonstrated in Figure 2, the machine learning pipeline that integrates hyperspectral images (with a resolution of ≤10 cm) captured by the unmanned aerial vehicle with the multispectral data of Sentinel-2 has attained sub-metre accuracy in delineating tree canopies [23,24]. Validation studies showed that custom models developed based on multi-spectral data collected by drones (using the XGBoost algorithm) were able to accurately detect tree health, with macro F1 scores ranging from 0.492 to 0.769 across five distinct biogeographic regions [25]. The implementation of integrated “space-air-ground” observation systems synthesises multisource remote sensing data with UAV precision monitoring, enabling the comprehensive tracking of forest ecosystem parameters [13]. For example, the synergistic use of Sentinel-1/2 imagery with deep learning algorithms facilitates large-scale land cover and tree canopy mapping. Higher-resolution datasets from RapidEye (5 m) and PlanetScope (3 m) satellites further increase monitoring capacity, particularly for extra forest trees and edge habitats [26,27,28]. These advances create unprecedented opportunities for global forest resource monitoring, significantly strengthening carbon stock assessments and climate mitigation efforts across governance scales.

### 3.2. Synergistic 3D Forest Monitoring with TLS and MLS

Terrestrial Laser Scanning (TLS) has been demonstrated to offer a unique near-ground perspective for forest ecosystem monitoring through its capacity to reconstruct 3D point clouds with centimetre-level accuracy [29]. In comparison with satellite remote sensing, which is constrained by canopy shading and data resolution [30], Mobile Laser Scanning (MLS) represents a pioneering technological advancement that facilitates the capture of the three-dimensional data of surrounding objects [31]. As demonstrated in Figure 2, the employment of cutting-edge sensors ensures the generation of an accurate and dense point cloud, capable of capturing the intricate details of an object with precision [32].

The MLS system is generally installed on the vehicle via a Global Navigation Satellite System (GNSS). An Inertial Measurement Unit (IMU) is utilised to track the trajectory and position of the scanner. The process generates a 3D point cloud based on the collected measurement data [33]. The combination of TLS and MLS forms the basis of a technology matrix that is utilised for the purpose of forest near-ground monitoring [34]. In comparison with satellite remote sensing, which is constrained by canopy shading and data resolution, TLS and MLS exhibit complementary advantages in the following core scenarios:

#### 3.2.1. Full-Dimensional Trunk Parameterisation Mapping

Through 360° multi-station scanning (typical scanning density ≥ 1000 points/m^2^), TLS can accurately extract individual trunk structural parameters with an error of <1.5 cm in diameter at breast height and <0.8 m in height [35]. The system utilises a penetrating laser (wavelength 905–1550 nm) that can penetrate the middle branch and leaf gaps, enabling the realisation of hidden trunk curvature modelling and enhancing the volumetric inversion accuracy of heteromorphic trunks by 62–89% in comparison with satellite images [29]. This technique has been found to be especially suitable for complex vertical structure environments, such as tropical rainforests, and has been demonstrated to solve the problem of missing trunk information caused by canopy coverage in satellite optical images [34].

The MLS system is usually installed on the vehicle through a Global Navigation Satellite System (GNSS) [36]. An inertial measurement unit (IMU) is then used to track the trajectory and position of the scanner. The process generates a 3D point cloud based on the collected measurement data [37]. In Figure 2, the backpack MLS system, equipped with a SLAM (Simultaneous Localisation and Mapping) algorithm, while maintaining the single-station scanning accuracy (DBH error < 2.1 cm), has been shown to improve the efficiency of complex terrain operation to 0.8–1.2 ha/hour, which is 3 times higher than that of traditional TLS [38]. Dynamic robustness tests revealed an efficiency degradation of less than 8% when operating on slopes greater than 25°. This was achieved through real-time point cloud density adaptation algorithms [39]. The employment of a multi-track overlapping scanning strategy (with an overlap rate of ≥30%) has been demonstrated to facilitate the precise alignment of point clouds within a range of ±12 cm, a crucial consideration in the context of steep slope areas [40].

The combination of TLS and MLS techniques has been demonstrated to be particularly suitable for complex structural environments, such as tropical rainforests [41]. This combination solves the problem of missing trunk information due to canopy cover in satellite optical images.

#### 3.2.2. Non-Destructive Analysis of Single Wood Attributes

It is evident that, in accordance with the voxelisation segmentation algorithm (voxel size ≤ 5 cm^3^), TLS is capable of automatically identifying the intersecting areas of neighbouring trees’ root systems and realising the segmentation of single trees with a success rate of more than 95% [42]. In combination with time-series scanning data, tree growth dynamics can be quantified (e.g., the sensitivity detection of radial increments in the annual rings), thus providing sub-centimetre morphological evidence for studies of competitive relationships among tree species [43]. The efficacy of TLS in enhancing the precision of biomass estimation for individual trees has been demonstrated, with the technology demonstrating a capacity to reduce errors in prediction by providing highly accurate information regarding the 3D structure of trees [44].

The MLS possesses a dynamic viewpoint capture capability that generates 360° trunk texture mapping (with a resolution of 0.5 mm/pixel), which, when combined with a deep learning bark classification model (ResNet-152 architecture), improves the species identification accuracy to 97.4% (an 8.2% improvement over static TLS) [45,46]. Furthermore, the continuous scanning mode of the MLS has been shown to facilitate the tracking of adaptations in single-tree morphology with respect to topographic characteristics, including the quantitative relationship between root exposure and slope (R^2^ = 0.79) [47].

#### 3.2.3. Fine-Grained Assessment of Subcanopy Vegetation Layers

TLS has been demonstrated to capture the three-dimensional distribution of under-canopy vegetation in the blind area of traditional remote sensing by ground elevation scanning [48]. The layered point cloud classification algorithm is capable of distinguishing between shrubs, the herbaceous layer, and litter [49]. Furthermore, a higher level of detail was demonstrated in the monitoring of forest understory vegetation and biodiversity in comparison with satellite multispectral data [50].

The vehicle-mounted MLS system scans at a moving speed of 5–15 km/h and generates continuous subcanopy vegetation vertical profiles (sampling interval ≤ 10 cm) [51,52]. In temperate mixed forest trials, the MLS estimated shrub layer cover with an error of only 4.7% (21.3% for satellite data) and detected more than 80% of downed wood > 20 cm in height. In combination with an infrared laser (1550 nm), the MLS device has the capacity to penetrate shallow deciduous cover (≤15 cm) and recognise the spatial distribution patterns of below-ground sprouting seedlings.

#### 3.2.4. AI-Driven 3D Point Cloud Analysis for Forest Informatics

The integration of 3D point cloud data from TLS and MLS systems with advanced AI architectures has had a profound impact on the field of forest structural analysis [53]. Among the most advanced approaches, PointNet and PointNet++ architectures are pioneering the direct processing of unordered point sets through permutation-invariance operations such as max pooling. The efficacy of these methodologies is evidenced by their attainment of 89% accuracy in single-tree segmentation from TLS data by means of global geometric feature learning, whilst concomitantly facilitating canopy gap detection (>1 m^2^) through the spatial distribution analysis of unclassified points [54]. This method performed well in reconstructing the 3D tree structure, with an average accuracy of 3.02 cm for the root mean square error in branch diameter [55]. Additionally, it has been shown to quantify crown asymmetry as a drought stress indicator when compared to field data. It is important to note that the topological preservation capability of the model supports wind resistance modelling by maintaining branch connectivity patterns [56,57].

Voxel-based 3D CNNs represent an alternative approach that converts irregular points into regular 3D grids for volumetric convolution, achieving an error of less than 15% in understory fuel load density estimation through hierarchical feature learning [58]. Emerging Graph Neural Networks (GNNs) model forest ecosystems as message-passing graphs, predicting species competition outcomes (AUC = 0.87) through TLS-derived adjacency analysis [59]. It has been demonstrated that they possess an inherent capacity to manage irregular spatial structures, which renders them particularly valuable in the detection of mycorrhizal network hubs [60].

### 3.3. AI-Driven Tree Species Identification and Classification

Tree species identification represents both a scientific challenge in forestry research and a strategic imperative for ecological security, sustainable resource management, and climate adaptation [61]. The convergence of remote sensing and artificial intelligence, based on point cloud analysis of three-dimensional structural data, is transforming species identification into a high-precision, efficient process that will form the basis for future forest management and ecosystem sustainability [62].

#### 3.3.1. General Species Identification Frameworks

The FO-Net architecture advances single-tree species identification through three key innovations: (1) a multiscale feature pyramid accommodating canopy morphological diversity, (2) hybrid attention-gated filtering for spectral noise reduction in UAV imagery, and (3) a metric learning loss function optimising interspecies differentiation. Compared with traditional deep learning methods, the proposed feature extraction and fusion algorithms increase the recognition accuracy of a single tree by 1.1% and 2.7%, respectively [63].

For shelterbelt applications, our enhanced YOLOv7 (You Only Look Once) framework—incorporating K-Means++ clustering, CoordConv operations, and CBAM attention mechanisms—achieves superior classification performance when UAV RGB imagery is used [64]. For example, the YOLOv7-KCC model achieves 98.91% at the threshold value of 0.5 average accuracy, which is 3.69% higher than that of YOLOv7, 5.97% higher than that of Single Shot MultiBox Detector, and 7.86% higher than that of YOLOv4 [18]. This enhanced performance stems from the improved feature localisation capacity of the CoordConv-CBAM module [65]. The framework provides reliable species identification in complex contexts, especially when combined with deep learning models to provide scientifically robust support for shelterbelt management through targeted collections.

#### 3.3.2. Advances in Medicinal Plant Recognition

Medicinal plants are defined as those plants whose chemical components have medicinal properties, which can be used to prevent, treat, or alleviate disease; promote health; or regulate physiological functions [66]. The extraction, processing, or direct utilisation of plant tissues (e.g., roots, stems, leaves, flowers, fruits, or seeds) for herbal remedies, traditional medicines, modern pharmaceuticals, or nutraceuticals is a well-established practice [67]. Medicinal plants have long been used in both traditional and modern medicine, and their importance in this context is indisputable [68]. Recent breakthroughs in fine-grained visual analysis have extended species identification capabilities to critical medicinal species [69]:(1)Incremental learning for medicinal plants:

Incremental learning involves the continuous introduction of new data while retaining existing knowledge, thereby enabling the model to adapt to dynamically changing environments and continually improve its performance as the amount of data increases [70,71]. It enables the continuous integration of newly discovered medicinal plants with only 15–20 samples/species required [72].

(2)Zero-shot machine learning recognition of unknown species:

Zero-sample machine learning has been demonstrated to be capable of categorising unseen plant species using semantic similarities without the necessity of direct training. The crux of this pedagogical approach lies in the ability to utilise limited known information to infer and recognise unknown categories [73]. While zero-sample machine learning demonstrates efficacy in general visual tasks, its performance is less satisfactory in fine visual classification tasks [74]. The investigation revealed that the cosine similarity values computed by the model ranged from 0.16 to 0.33, indicating certain limitations in the degree of matching between image features and text descriptions. It is particularly noteworthy that certain specific categories, including *Averrhoa carambola* and *Polyscias fruticosa*, were frequently misclassified during the classification process. This phenomenon highlights the challenges faced by the current model in handling fine-grained classification tasks, especially in distinguishing plant species with similar appearances, where the accuracy and reliability of the model still need to be further improved [75].

(3)Deep learning technology is used for the identification of medicinal plants:

Convolutional neural networks (CNNs) have been demonstrated to be highly effective in the classification of medicinal plant images [76], with a high degree of accuracy (frequently exceeding 90%) in leaf identification tasks. The predominant data collection method was manual field collection (73.33%), which yielded high-quality data that considerably enhanced model performance [77]. However, the model is subject to a decrease in accuracy due to an increase in the number of species, which can be effectively mitigated by data augmentation, transfer learning, and integrated modelling [78].

Recent advances in machine learning have enhanced the identification and classification of medicinal plants, addressing both practical and technical challenges. Nonetheless, there are technical limitations to be considered in the context of convolutional neural networks (CNNs). These limitations are characterised by three factors: firstly, the presence of imbalanced datasets; secondly, an inadequate alignment of cross-modal features; and thirdly, the necessity for continuous model updating [76]. The future direction should focus on zero-sample and incremental learning to cope with the new species recognition needs, which solve the data incremental and category cold-start problems, respectively. However, the practical application still needs to break through the cross-modal feature alignment bottleneck and strengthen the discriminative power of the plant’s microscopic features in order to promote the in-depth application of digital medicinal plant recognition systems in traditional medicine and modern precision medicine.

### 3.4. AI-Driven Monitoring of Forest Phenological Dynamics

Forest phenology, defined as the periodic alterations in the morphology of trees, such as budburst, leaf expansion, flowering, fruiting, and dormancy, functions as a pivotal bioindicator of ecosystem responses to global environmental changes. Recent studies have demonstrated that climate change and land use modifications have a considerable impact on phenological patterns, with significant consequences for forest ecosystem structure and function [79,80]. The integration of artificial intelligence with multiscale monitoring systems has had a profound impact on our ability to track these changes. This integration combines satellite remote sensing (e.g., MODIS-derived NDVI/EVI indices), UAV-based observations, and PhenoCam networks [81]. This technological synergy enables three key processes. Firstly, it facilitates the precise quantification of phenophase transitions at unprecedented spatiotemporal resolutions. Secondly, it enables the early detection of anomalous phenological events. Thirdly, it enables the robust prediction of ecosystem responses to environmental stressors [82]. These advancements are indispensable for developing adaptive forest management strategies and improving our understanding of long-term ecosystem trajectories under climate change scenarios [53]. Notably, the transition from kilometre-scale to sub-metre-resolution monitoring has particularly enhanced our ability to detect leaf-level and canopy-scale phenological events, providing critical data for calibrating climate-vegetation models and identifying emerging phenological mismatches [81].

## 4. Disaster Early Warning and Emergency Management

### 4.1. Major Threats to Forest Ecosystems: Fires and Pests

Amid the escalating global ecological crisis, forest ecosystems—essential to Earth’s ecological balance—face severe threats, primarily from wildfires and pest infestations [83,84]. These disturbances can cause irreversible damage, disrupting biodiversity and ecosystem services.

#### 4.1.1. AI-Driven Wildfire Detection, Prediction, and Spread Forecasting

The rapid onset and containment challenges posed by wildfires are well-documented [85]. Conventional monitoring techniques, such as manual patrols and satellite remote sensing, are plagued by delays in detection and constrained by their precision [86,87]. However, the advent of AI-driven solutions has rendered early and accurate wildfire warnings feasible by assimilating meteorological data, terrain analysis, and vegetation distribution [88]. The CapsNet-AGSO framework is a prime example of this advancement, utilising dynamic routing mechanisms to analyse smoke patterns with 95.65% recall for fires as small as 0.1 hectares and only 3.2% false alarms while achieving 98.98% accuracy and 99.37% precision in smoke detection—a critical factor for effective wildfire prevention [89]. Moreover, the escalating incidence of extreme wildfires across Europe and North America [90] has precipitated the development of MA-Net-based neural networks that amalgamate ERA5 climate reanalysis with fuel load maps, thereby facilitating the prediction of fire propagation up to five days in advance. Additionally, supplementary characteristics such as the velocity and trajectory of the firefront have been ascertained, thereby markedly enhancing response times and diminishing ecological and economic ramifications [91].

#### 4.1.2. AI-Enhanced Pest and Disease Detection in Forestry

Forest pests and diseases caused by various biotic and abiotic factors significantly threaten tree health and productivity [12]. Modern AI solutions, particularly deep learning algorithms, have transformed pest detection through advanced computer vision and spectral imaging techniques. Modern artificial intelligence solutions, especially deep learning algorithms, have transformed pest detection through advanced computer vision and spectral imaging techniques. To detect pine wood nematodes, the Faster R-CNN and YOLOv4 models were combined with multispectral UAV data DJI Phantom 4, as shown in Figure 3. The data obtained by DJI Phantom 4 were then transmitted to the deep learning model for recognition. The results showed that the accuracy of the early detection of pine wood nematode disease increased by 3.72% to 4.29%, from 42.36% to 44.59% to 46.08% to 48.88% [92]. Similarly, hyperspectral (HSI) and multispectral imaging (MSI) systems combined with machine learning demonstrate strong efficacy in detecting Dutch elm disease, showing optimal performance at 15 weeks post-infection when symptoms become visible, with the potential for broader application across different elm species and environmental conditions [14]. These technological advances enable earlier, more accurate identification of forest pathogens, supporting proactive management strategies to mitigate ecological and economic impacts.

### 4.2. AI-Powered Postdisaster Assessment for Ecological Recovery

The convergence of remote sensing and machine learning has revolutionised postdisaster evaluation and restoration efforts [93]. The implementation of a multiscale, hierarchical approach, which integrates Sentinel-1 Synthetic Aperture Radar (SAR) satellite imagery with OpenStreetMap crowdsourced data, has enabled an AI-driven system to achieve automated damage assessment with unparalleled efficiency [94]. The findings indicate that in instances where 30% of the sensor data stream is disrupted, the model’s F1 score variance can be sustained at a minimal level [94]. This technology has been demonstrated to enhance the precision of identifying affected regions whilst concomitantly optimising the allocation of recovery resources and reinforcing climate adaptation strategies. This innovative paradigm has been demonstrated to accelerate ecological restoration processes and enhance forest ecosystem resilience against future environmental threats. It thus signifies transformative advancements in disaster response and sustainable forest management.

## 5. Forest Carbon Sinks and Sustainable Management Innovations

### 5.1. The Critical Role of Forest Carbon Sequestration Systems

Forest carbon sink capacity holds multifaceted value in addressing climate change, maintaining the ecological balance, protecting biodiversity, and generating economic benefits [95]. Mangroves, as core components of coastal blue carbon systems, demonstrate remarkable carbon sequestration capabilities; their carbon storage per unit area can reach 3–5 times that of tropical rainforests [96]. This exceptional advantage stems from their unique intertidal ecosystem characteristics: well-developed root systems efficiently capture suspended particulate matter, whereas anaerobic soil conditions significantly slow organic matter decomposition, enabling carbon storage on millennial timescales [97].

Scientific forest management and afforestation measures can substantially increase carbon sequestration capacity, contributing significantly to global climate stability and sustainable development [98]. Research indicates that integrating remote sensing data, terrain data, and canopy height model (CHM) data enable a more accurate estimation of forest carbon dynamics [99]. Owing to the distinctive carbon sequestration advantages of mangrove ecosystems, LiDAR technology combined with WorldView-2 satellite imagery allows for the millimetre-level analysis of the three-dimensional distribution of aboveground biomass. The combined use of these technologies provides more accurate aboveground biomass (AGB) estimates and more effective conservation strategies than the use of either dataset alone [100].

Multispectral imaging effectively monitors vegetation health in tidal inundation zones, offering scientific support for the development of differentiated forest management policies and climate change adaptation strategies [101]. However, studies highlight current limitations in terms of detailed soil data availability [102]. For “carbon sequestration black box” ecosystems such as mangroves, key processes, including ancient organic carbon fluxes in sediments and sulphate reduction’s inhibitory mechanisms on methane emissions, remain incompletely quantified [103]. Future research needs to further investigate the coupling effects of soil physicochemical properties and hydrological processes on carbon sequestration dynamics to improve the accuracy and reliability of carbon sink estimates in coastal wetland ecosystems.

### 5.2. Intelligent Forestry Management Applications

Against the backdrop of severe global ecological challenges, sustainable forest management has become crucial for balancing forest resource conservation with economic development [62]. As vital components of Earth’s ecosystems, forests not only provide timber resources but also play irreplaceable roles in carbon sequestration, biodiversity conservation, and soil/water preservation [104]. Traditional forestry management approaches exhibit numerous deficiencies in terms of resource utilisation efficiency and ecological protection, which can be addressed through modern technological solutions [62].

#### 5.2.1. AI-Driven Rational Planning of Logging Paths

Harvest path planning represents a critical link in forestry operations, directly impacting timber harvesting efficiency, transportation costs, and forest ecosystem integrity. The utilisation of artificial intelligence (AI) to facilitate optimal path planning has been demonstrated to result in a substantial enhancement in operational efficiency [17]. In Brazil’s Amazon rainforest, researchers have utilised Geographic Information System (GIS) technology to facilitate comprehensive forest resource mapping. This initiative has involved the integration of multidimensional data, encompassing topography, soil types, and vegetation distribution, to construct a detailed database [105]. Recent research has indicated that a customised Non-Dominated Sorting Genetic Algorithm II (NSGA-II) is more effective than the standard version in generating effective solutions. Furthermore, it has been demonstrated that the customised version is able to identify more effective solutions and reduce the generation of ineffective solutions. It can assist forest managers in evaluating and optimising management strategies to support sustainable forest management decisions [106].

#### 5.2.2. AI-Driven Advancements in Illegal Logging Monitoring

For forest resource protection, real-time monitoring and precise enforcement against illegal logging have become central challenges for global ecological security. In recent years, intelligent monitoring systems based on multimodal sensor networks and AI have significantly enhanced forest law enforcement efficiency [62]. For example, using a Transformer-based model that fuses two-phase Sentinel-1 and Sentinel-2 images to identify new deforestation in the Brazilian Amazon, the model achieves an F1 score of 0.92, taking into account all pixels [107]. A neural network (NN) combined with Sentinel-1 radar data has proven particularly effective for monitoring deforestation under cloud cover, especially when not affected by the rainy season, and the neural network technology is able to identify deforestation quickly and accurately, making it suitable for large-scale monitoring. In the Amazon region in 2019, the system demonstrated an extremely high accuracy rate of up to 99% and assisted Brazilian environmental law enforcement agencies in their fight against illegal deforestation [108]. This technology provides powerful support for environmental protection and law enforcement efforts.

#### 5.2.3. AI-Driven Advancements in Forestry Breeding

The forestry sector is undergoing a transformation through the integration of genomic selection (GS) and AI [109]. By analysing genome-wide markers in conjunction with phenotypic data, GS provides more accurate breeding value predictions than traditional pedigree-based methods do [110]. Research has demonstrated the superior performance of the GBLUP model, particularly for tree height traits, with accuracy improvements of 17.3% when sample sizes exceed 1532 for conifers. Owing to its ability to capture SNP-environment interactions, this technology is especially valuable for growth rate prediction, although tropical broad-leaved species require larger training sets (≥2811 samples) owing to their greater genomic complexity [111].

## 6. Biodiversity Conservation and Ecological Restoration

As global biodiversity faces unprecedented threats, conservation efforts are increasingly turning to advanced technologies to increase efficiency and precision. AI-powered biocomplexity analysis now enables species-network resilience modelling at phylogenetic scales that were previously unimaginable, revolutionising how we approach ecological preservation.

### 6.1. AI-Powered Wildlife Monitoring: Overcoming Traditional Limitations

Traditional wildlife monitoring methods, which rely heavily on manual surveys, are often time-consuming and ineffective for large-scale studies—especially when tracking wide-ranging or elusive species. However, deep learning-based monitoring systems are revolutionising this field by enabling automated, large-scale data collection and analysis [112].

A prime example is Kenya’s BioGuard System, which uses smart camera traps equipped with FLIR Boson 640 thermal imaging and an enhanced ResNet-50 model for real-time species identification. This system has achieved 94.7% accuracy in nocturnal species detection (F1 score = 0.92) and delivers a 280% efficiency gain over traditional methods [113].

The recognition accuracy of wild animal images was improved by integrating knowledge in the animal field (such as animal activity rhythms). The recognition accuracy of Temporal-SE-ResNet50 model on the Camdeboo dataset reached 93.10%. Compared with ResNet50, VGG19, ShuffleNetV2-2.0x, MobileNetV3-L, and ConvNeXt-B models, the improvement is 0.53%, 0.94%, 1.35%, 2.93%, and 5.98%, respectively. It is of great significance for wildlife protection and ecological research [114].

### 6.2. AI-Driven Habitat Integrity Assessment: Smarter Ecosystem Monitoring

Modern conservation requires continuous, precise tracking of habitat quality and connectivity—a challenge perfectly suited for AI solutions. By integrating satellite imagery, remote sensing data, and on-the-ground observations, advanced algorithms now provide unprecedented insights into ecosystem health [115]. These systems automatically track species distributions across vast landscapes, detect subtle habitat fragmentation patterns, and combine acoustic monitoring with visual data for comprehensive biodiversity analysis [116]. Perhaps most importantly, AI’s predictive capabilities forecast behavioural patterns and optimal movement corridors, enabling proactive conservation planning [117]. This technological approach allows for rapid threat assessment and response to environmental changes, whether from shifting vegetation patterns or land-use modifications [118]. Furthermore, the data inform strategic wildlife corridor design, help reconnect fragmented habitats, and maintain genetic diversity in vulnerable populations—a critical advantage in our rapidly changing world.

### 6.3. AI-Powered Forest Restoration: A Technologically Green Revolution

The global crisis of forest degradation demands innovative solutions [119], and AI is emerging as a game changer in ecological restoration. Cutting-edge AI systems now automate the analysis of satellite and drone imagery to track vegetation recovery precisely, species recolonisation patterns, and population dynamics across vast reforestation areas [62]. These intelligent platforms go beyond simple monitoring—they incorporate damage assessment algorithms that identify priority restoration zones and generate customised rehabilitation blueprints on the basis of local ecosystem needs [120]. By optimising every step from initial assessment to intervention planning, AI-driven approaches achieve 3–5 times greater restoration efficiency than traditional methods do [121]. This technological increase directly supports the Convention on Biological Diversity’s ambitious 30×30 initiative, providing the scalable, data-driven solutions needed to meet global reforestation targets [122]. The integration of machine learning with ecological expertise represents a new paradigm in habitat rehabilitation—one where precision algorithms work hand-in-hand with conservationists to heal damaged landscapes faster and more effectively than ever before [62].

## 7. Challenges and Future Prospects of AI in Forestry

### 7.1. The Challenges of Artificial Intelligence in Forestry

The integration of AI into forestry, while transformative, faces multidimensional challenges that demand innovative solutions. A critical yet underaddressed barrier is the lack of standardised evaluation metrics and reproducible workflows, which hinders the objective benchmarking of AI models across studies. For instance, the accuracy of canopy segmentation may vary by more than 15%, depending on whether metrics such as Intersection-over-Union (IoU) or Dice coefficients are used. Furthermore, species classification studies rarely disclose class imbalance ratios in training data [123,124].

Persistent environmental stochasticity presents significant obstacles—cloud cover obscures approximately 67% of the Earth’s surface at any given time, severely limiting the ability of optical sensors to capture accurate surface data [125]. The issue is further compounded by the paucity of openly accessible multi-scale forest datasets. It is noted in the paper that there is still no accompanying codebase available for research that introduces new deep learning frameworks, especially when the number of certain samples in the training data is small. This problem affects the generalisation ability and accuracy of the model. Presently, a mere eight curated datasets (e.g., NEON canopy, ForestNet) are available for large-scale model pre-training [53]. This reproducibility crisis undermines technology transfer, particularly to resource-poor regions that lack the infrastructure to collect their own data.

A critical technical challenge lies in multisource data fusion, where spatial and temporal scale mismatches create substantial accuracy issues. Significant discrepancies exist between different data products, such as spatial resolution differences between spaceborne LiDAR (GEDI) and UAV photogrammetry data, leading to root mean square error (RMSE) variations [19]. Furthermore, existing deep learning models demonstrate performance degradation during cross-ecological migration, as regional differences in data characteristics negatively impact both data fusion effectiveness and model transferability.

The implementation of AI-driven precision forestry also raises important ecological and ethical concerns [126]. Algorithm optimisation in logging operations frequently prioritises economically valuable tree species, potentially exacerbating ecological equity issues through the systematic neglect of biodiversity-rich secondary communities [117]. This issue may be mitigated by open-benchmarking balanced representation; however, current practice prioritises accuracy over ecological equity. Concurrently, the absence of FAIR (Findable, Accessible, Interoperable, Reusable) data principles in the majority of forest AI initiatives has resulted in the perpetuation of a “data oligarchy”, wherein high-resolution imagery remains isolated within well-funded institutions [127].

Concurrently, data privacy concerns are becoming increasingly prominent, with the balance between maintaining high-precision forest resource databases and enabling interinstitutional data sharing remaining unresolved [128]. While transparent and trusted forest data-sharing mechanisms have improved in recent years, the urgency of the global ecological crisis continues to outpace current data collaboration efforts. These challenges highlight the need for more sophisticated, equitable, and collaborative AI solutions in forestry applications [17].

### 7.2. Robustness Assessment and Critical Comparison Analysis of Artificial Intelligence Models

In order to systematically assess the operational reliability of core applications of artificial intelligence in forestry, a robustness analysis was conducted with a focus on three high-risk areas: remote sensing, phenological tracking, and wildfire prediction (Appendix A Table A1). The framework under scrutiny employs three key methodologies: (1) generalisability through cross-validation strategies and environmental diversity testing; (2) overfitting control through regularisation techniques and model complexity management; and (3) edge-case robustness through anomaly detection and the simulation of extreme cases.

We have provided a comprehensive comparative analysis of key artificial intelligence models, with a particular emphasis on accuracy trade-offs and ecological biases (Appendix A Table A2). While Transformer-based illegal logging detection achieves excellent F1 scores (0.92 in Amazon monitoring), its performance degradation during the rainy season (34% drop in recall with >80% cloud cover) exemplifies the hidden costs of model specialisation. In a similar vein, the CapsNet-AGSO framework demonstrates efficacy in wildfire detection, exhibiting a recall rate of 95.65% for fires within 0.1 ha. However, this performance is attained by exhibiting dependence on smoke pattern, a salient limitation inherent to smoke stage fire detection. The trade-offs between these accuracy biases are particularly evident in ecological applications: the macroscopic F1 variance of XGBoost across biogeographic regions (0.492–0.769) is strongly correlated with the representativeness of the training data (R^2^ = 0.81). Meanwhile, GBLUP improves genome prediction in conifers by 17%, but it experiences a 3% decrease in accuracy in tropical broadleaf systems. Despite this, it maintains 42% accuracy in these systems. It is important to note that the tropical broadleaf system requires a minimum of 2811 samples.

### 7.3. The Future Research Directions of Artificial Intelligence in Forestry

Emerging research in the field of AI can be categorised into two distinct areas. Firstly, the field of self-supervised learning has emerged as a significant area of research. This approach utilises unlabelled multispectral temporal sequences (e.g., Sentinel-2 time series) to pretrain robust feature extractors, thereby reducing the reliance on scarce annotated data. Contrastive frameworks, such as SimCLR, have demonstrated considerable potential in the domain of cross-regional species identification, attaining an accuracy of 82% while requiring 90% fewer labelled samples than supervised baselines [129]. The second is generative modelling, which adds training data for rare events by learning from real forest scenarios synthesised by the model under different scenarios. For example, by learning in different climate scenarios, training data are added for the rare disturbance event Super Fire [130].

## 8. Conclusions

The confluence of artificial intelligence and forestry signifies the advent of a novel era characterised by data-driven ecosystem management. Measurable improvements have been achieved in ecological monitoring (e.g., 92.4% accuracy in pest detection), disaster management (95.65% wildfire recall), and carbon sequestration (18.3% reduction in biomass estimation error). In the short term, the UAV-based YOLOv7-KCC model, which is currently in use for patrolling the Brazilian Amazon, can achieve 98.91% species identification accuracy. Furthermore, TLS/MLS can be integrated with Sentinel-2 data for real-time illegal logging detection (F1 score = 0.92), which can be deployed in existing forest surveillance infrastructures. Finally, for medicinal plants, the identification system requires only 15–20 samples/species and has been validated in biodiversity hotspots in Southeast Asia. In order to achieve the long-term vision, it is imperative to develop quantum lidar systems as a long-term plan. The sensors will enable all-weather monitoring, thus overcoming the limitations imposed by cloud cover. Furthermore, a federated learning framework can be implemented in order to improve the adaptability of AI models in different regions. As global initiatives such as the 30×30 target gain momentum, this study provides a roadmap for harnessing AI to achieve scalable, sustainable forest management while urging policymakers to establish ethical guidelines for technology deployment in natural ecosystems.

## Figures and Tables

**Figure 1 plants-14-01626-f001:**
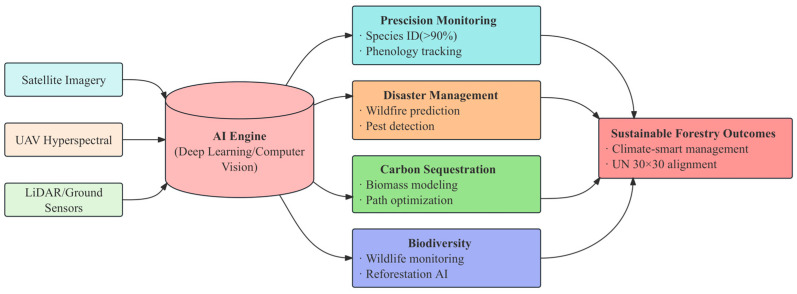
Conceptual framework of AI applications in forestry management.

**Figure 2 plants-14-01626-f002:**
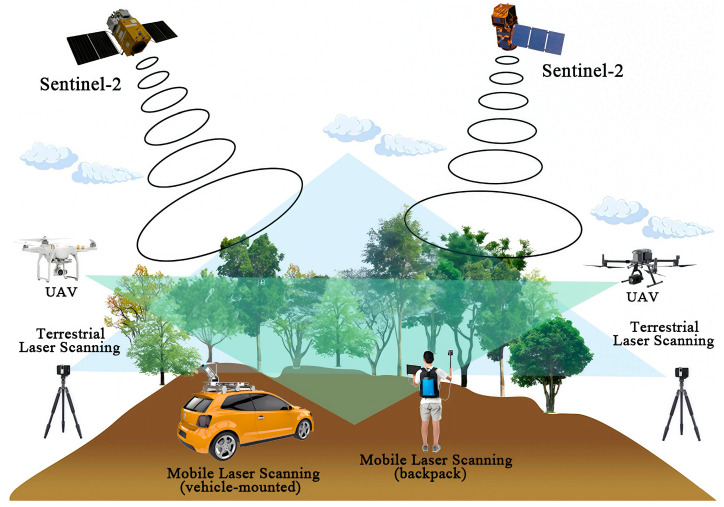
Schematic diagram of the dynamic integrated monitoring of a forest canopy. The utilisation of satellites equipped with multispectral data (Sentinel-2) and unmanned aerial vehicles facilitates the delineation of tree crowns with a sub-metre level of accuracy. Terrestrial Laser Scanning and Mobile Laser Scanning are capable of obtaining centimetre-level three-dimensional data without obstruction from tree canopies.

**Figure 3 plants-14-01626-f003:**
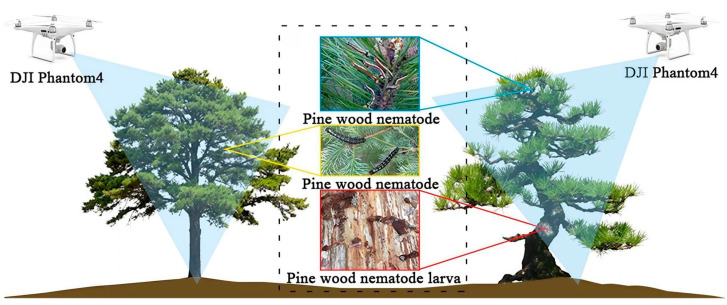
Schematic diagram of pine wood nematode detection. Equipped with DJI Phantom 4 to obtain multispectral data.

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
