# Peer review of "AI-Powered Plant Science: Transforming Forestry Monitoring, Disease Prediction, and Climate Adaptation"

_plants, 2025, doi:10.3390/plants14111626_

Round 1
Reviewer 1 Report
Comments and Suggestions for Authors
Please see attached.

Reviewer 2 Report
Comments and Suggestions for Authors
The text is short but addresses the main potentials of AI implementation in forestry and environmental monitoring. Overall, I think it is well written, and I don’t have any major comments. However, I believe some sections could explore the potential of AI in more depth.
One noticeable gap is the lack of discussion on Terrestrial Laser Scanning (TLS) and Mobile Laser Scanning (MLS)—technologies that are among the most challenging yet valuable in forest inventory. I suggest adding a section detailing the role and advantages of terrestrial LiDAR in forest monitoring, particularly for stem mapping, individual tree analysis, and under-canopy vegetation assessment, which effectively complement satellite-based approaches. There is a big potential of AI in this area.
Additionally, I recommend including a subsection summarizing AI methods specific to 3D point cloud data, especially for readers interested in the technical foundations of forest informatics. This could include a review of state-of-the-art AI architectures such as PointNet, PointNet++, DGCNN, CNN-based voxelization methods or Graph Neural Networks.
In the context of AI technologies, one of the major challenges lies in data standardization and reproducibility. It would be valuable to emphasize the current lack of standardized evaluation metrics, open-access datasets and code, and the broader issue of research reproducibility.
Lastly, your text could be further enriched by addressing future research directions in forestry AI, including self-supervised learning, and generative modeling, which represent promising but underexplored areas.
Reviewer 3 Report
Comments and Suggestions for Authors
Dear Authors,
I have reviewed the paper titled: “AI-Powered Plant Science: Transforming Forestry Monitoring, Disease Prediction, and Climate Adaptation". In my opinion, the aims of the paper are germane with “Plants” journal topic, however in the present form, this paper has some important flaws as also in part reported below:
- The review lacks of a clear structure and looks like a compendium of possible applications of AI in forestry without a specified background.
- The lack of a M&M section jeopardises the possibility of understanding what the authors wanted to do, why and how.
- There is a general great confusion and it is hard for the reader to find a flow of thoughts and to understand the aim of the review and to what kind of audience it is addressed.
- Furthermore, there is a very recent review dealing with a similar topic https://doi.org/10.1007/s40725-024-00231-7
In the present form the contribution of this paper to the scientific knowledge is low. I understand the difficult work done, but as a reviewer it is my duty to highlight the gaps in order to improve the research approach and its presentation to the international scientific community. The manuscript does not satisfy the quality requirements for publication in a scientific journal, and it is much closer to a technical report than to a scientific paper.
Round 2
Reviewer 1 Report
Comments and Suggestions for Authors
Reasonable revisions
Author Response
Comments 1: Reasonable revisions
Response 1: Thank you for acknowledging the revisions.
Reviewer 3 Report
Comments and Suggestions for Authors
Dear Authors,
I have reviewed for the second time the paper titled: “AI-Powered Plant Science: Transforming Forestry Monitoring, Disease Prediction, and Climate Adaptation". In my opinion, the aims of the paper are germane with “Plants” journal topic, however in the present form, this paper has still some important flaws. In detail, I appreciate your efforts in improving your manuscript, however, its overall clarity and quality is still far from being satisfactory. Here the major comments:
- the abstract is very confusing and you are reporting specific numerical results without giving a clear background, the readers do not understand what is your scope and why you are presenting such results and not others.
- the introduction, despite the efforts in improving it, is still confusing. Probably, authors took a too wide topic, there is too much literature on AI applications in forestry for being condensation efficiently in only one review, the risk is to briefly touch each topic without enough detail not bringing anything tangible to the literature. I sincerely suggest to select 2 to 5 AI applications with a logical order and connection, for instance those related to biodiversity, those related to pest management, etc... and build the review by focusing on these specific applications, giving clear information to the readers about why these are important, what is the state of the art and how and how much AI can advance our knowledge. This is completely missing in the review, but it is normal because there are too many data and inputs to create a logical flow.
- In M&M you should clearly indicate the number of papers selected for the review, I am pretty sure it is almost impossible to collect and analyse all of them, practically each paper dealing with remote sensing in forestry uses AI algorithms and I am wondering how it is possible to merge all of them in a single review.
- this lack of clarity reflects on the other sections of the review, you should rethink your topics, describe its importance and then show specific applications and advantages of AI, giving clear research directions and management suggestions. It is just impossible to do this with all your enormous amount of information, 500 pages or more would be needed...
